# Towards Predicting the Sequential Appearance of Zeolitic Imidazolate Frameworks Synthesized by Mechanochemistry

**DOI:** 10.3390/molecules27061946

**Published:** 2022-03-17

**Authors:** Mohamed Cherif, Gaixia Zhang, Yang Gao, Shuhui Sun, François Vidal

**Affiliations:** Centre Énergie, Matériaux, Télécommunications, Institut National de la Recherche Scientifique, 1650 Bd. Lionel-Boulet, Varennes, QC J3X 1P7, Canada; mohamed.cherif@inrs.ca (M.C.); gaixia.zhang@inrs.ca (G.Z.); yang.gao@inrs.ca (Y.G.); shuhui.sun@inrs.ca (S.S.)

**Keywords:** mechanochemistry, metal organic frameworks, zeolitic imidazolate frameworks, density functional theory, van der Waals interactions, molecular dynamics, hydrostatic pressure

## Abstract

We use computational materials methods to study the sequential appearance of zinc-based zeolitic imidazolate frameworks (ZIFs) generated in the mechanochemical conversion process. We consider nine ZIF topologies, namely RHO, ANA, QTZ, SOD, KAT, DIA, NEB, CAG and GIS, combined with the two ligands 2-methylimidazolate and 2-ethylimidazolate. Of the 18 combinations obtained, only six (three for each ligand) were actually observed during the mechanosynthesis process. Energy and porosity calculations based on density functional theory, in combination with the Ostwald rule of stages, were found to be insufficient to distinguish the experimentally observed ZIFs. We then show, using classical molecular dynamics, that only ZIFs withstanding quasi-hydrostatic pressure *P* ≥ 0.3 GPa without being destroyed were observed in the laboratory. This finding, along with the requirement that successive ZIFs be generated with decreasing porosity and/or energy, provides heuristic rules for predicting the sequences of mechanically generated ZIFs for the two ligands considered.

## 1. Introduction

Imidazolate-based zeolite structures (ZIFs) represent a subclass of metal-organic structures (MOFs). In ZIFs, each metal node is linked to four others by imidazolate groups via metal–nitrogen bonds to form porous, crystal structures with zeolite-like topologies [1]. MOFs, in general, and ZIFs, in particular, have attracted much attention due to their wide range of applications, including catalysis [2], CO_2_ capture [3], gas separation [4], chromatographic separations [5], sensing and drug delivery [6]. Among the techniques available to synthesize ZIFs, mechanochemical milling is an efficient, fast and cleaner alternative to conventional chemical syntheses [7,8,9,10,11,12,13]. Grinding involves synthesis from elemental constituents by impacting metal balls into a vessel. These balls typically have a mass on the order of 1 g and a diameter of a few mm [14]. It has been reported that, in liquid-assisted milling, the reaction between ZnO and the ligands 2-methylimidazolate (HMeIm) or 2-ethylimidazolate (HEtIm) (see Figure 1) leads to the generation of ZIF polymorphs incorporating the corresponding ligand following the reaction [15]:
ZnO + 2HMeIm or 2HEtIm → ZIF-Zn(HMeIm or HEtIm)_2_ + H_2_O(1)

During continuous grinding, different ZIF topologies are generated one after the other. For the HMeIm ligand, the mechanosynthesis process was shown to first yield SOD (ZIF-8), followed by an amorphous phase, then KAT (katsenite) and DIA (diamondoid) topologies:
HMeIm: SOD → amorphous → KAT → DIA(2)
while the HEtIm ligand produced, sequentially, the RHO (zeolite-ρ), ANA (ZIF-14) and QTZ (β-quartz) topologies:
HEtIm: RHO → ANA → QTZ(3)

It has been observed that ZIFs generated in these mechanochemical processes undergo a series of transformations into less energetic and more compact phases [15]. The amorphous phase following SOD-Zn(HMeIm)_2_ has not been characterized to our knowledge. The experimental data reported in [15] regarding the transformation enthalpies for the sequences (2) and (3) were supported by density functional theory (DFT) calculations including van der Waals (vdW) interactions. 

A question that naturally arises is why certain ZIF topologies are observed with one ligand and not with the other ligand in the mechanochemical synthesis process. In other words, why, for example, were RHO-Zn(HMeIm)_2_ and SOD-Zn(HEtIm)_2_ not observed in the laboratory? This work aims to shed some light on this complex issue [16]. To do so, we modeled the 12 ZIFs resulting from the combination of the six topologies RHO, ANA, QTZ, SOD, KAT and DIA of sequences (2) and (3) with the two ligands HMeIm and HEtIm and searched for criteria to identify the experimentally observed ZIFs and determine their order of appearance. In addition to these 12 ZIFs, we considered six hypothetical ZIFs formed by the three topologies, NEB, CAG (ZIF-4) and GIS (ZIF-6) with the same two ligands. In fact, the last three topologies with the 2-imidazolate ligand (HIm; Figure 1) were studied numerically in [17]. These six additional ZIFs were added to strengthen our findings.

In this work, we first determined the energies and porosity of the 18 ZIFs considered by means of DFT to show that the identification of the experimentally observed ZIFs cannot be achieved by considering only these two parameters. Then, considering that mechanochemistry involves high pressures exerted by the impact of the balls on the material covering the reactor walls, we turned to the calculation of an additional parameter of the ZIFs, namely the quasi-hydrostatic pressure breaking point (i.e., the pressure at which the crystal loses its integrity), using classical molecular dynamics (CMD) for infinite crystals and for clusters made of a few unit cells. The high pressures involved in the mechanosynthesis process could be easily highlighted. Indeed, if the ball lost most of its kinetic energy after an impact, the pressure could be estimated as the ratio of its kinetic energy to the compressed volume *V*:*P* ≈ ½ *m v*^2^
*V*^−1^(4)
where *m* is the mass of the ball, and *v* is its velocity. Realistic values of these parameters, *m* ≈ 1 g [14], *v* ≈ 10 m s^−1^ [18], *V* ≈ 0.1 mm depth × 1 mm^2^ area, lead to a pressure on the order of a tenth of a GPa. We identified this pressure breakpoint by an abrupt change in volume upon application of a quasi-hydrostatic pressure on the crystal. We use the prefix quasi because the structures were subject to pressure fluctuations in the CMD simulations, and the applied pressure varied in small steps of 1 ns, as explained in Section 2. We found that the experimentally observed ZIFs were those with the highest pressure breakpoint. This finding, combined with the requirement that the energy and/or porosity of the crystal decrease for each new ZIF generated, allowed us to predict the sequence of ZIFs produced by mechanochemistry.

## 2. Results

Calculation methods and definitions can be found in Section 3.

### 2.1. DFT Optimizations

Figure 2 shows the calculated total energies of the 18 ZIFs considered as a function of their packing index for the ligands HMeIm (Figure 2a) and HEtIm (Figure 2b). The energy of each structure was normalized to the number of Zn atoms per unit cell and plotted against that of the lowest energy polymorph for each ligand, namely DIA and QTZ for the HMeIm and HEtIm ligands, respectively. Two types of calculation are presented in this figure: S-DFT (standard DFT) with the PBE functional [19] and a van der Waals density functional (vdW-DF) with the functional described in [20], hereafter called BH. The results for the same structure are connected by line segment. The BH functional was selected for Figure 2 because it provided the smallest overall error on the cell volumes compared to the experimental values (see Table 1). Figure A1 of the Appendix A shows that the results for the six different vdW-DFs were similar. We first observe, in Figure 2, the general trend that the most porous structures had the highest relative energy. We also observe that vdW interactions increased the packing index compared to S-DFT. This effect was expected since vdW interactions exert a long-range attraction and, thus, bring the atoms closer together. The relative energies obtained, here, between the experimentally observed ZIFs were larger than those reported in [15]. However, the energy order of the experimentally observed ZIFs was the same as in [15] for both S-DFT and vdW-DF, viz:HMeIm: *E*[SOD] > *E*[KAT] > *E*[DIA](5)
HEtIm: *E*[RHO] > *E*[ANA] > *E*[QTZ](6)

Table 1 aims to support the accuracy of our calculations by comparing the unit cell volumes obtained by S-DFT and vdW-DF at 0 K and 300 K with those measured experimentally for the ZIFs of the sequences (2) and (3). The calculations at 300 K were performed by quantum molecular dynamics (QMD), as explained in Section 3. We observe that the vdW-DF calculations at 300 K were within 1% relative error of the measurements, (with the exception of QTZ for reasons that are not yet clear) and in better agreement with measurements than S-DFT. For comparison, the deviation of the calculated volumes by CMD at 300 K from measurements was between 3.2% and 4.6% (not shown in Table 1). The S-DFT and vdW-DF volumes at 300 K were consistently larger than those at 0 K, indicating that the thermal expansion coefficient was positive. We verified on the same six structures that this trend persisted at 10 K. The vdW-DF volumes at 300 K were closer to the measurements, which is consistent with the fact that the latter were made under normal temperature conditions. Table A1 in the Appendix A shows that the lattice parameters a, b and c and the angles α = (b, c), β = (c, a) and γ = (a, b), calculated with S-DFT and with vdW-DF at 0 K, were also consistent with the experimental results. We verified that the free energies at 300 K were almost the same as those at 0 K presented in Figure 2.

Mechanochemical structural transformations, in which the energy of successive structures decreases as they become more compact, have been linked to the Ostwald rule of stages (ORS) [15]. According to the ORS transformation pathway, ZIFs would first appear in the upper-left corner of Figure 2a or 2b and undergo a series of transformations each time to the next polymorph located at a lower energy and higher packing index. Therefore, according to Figure 2a, the ZIFs in the HMeIm sequence should be generated as (ZIFs in bold characters indicate those that were observed):
HMeIm (S-DFT and vdW-DF): RHO → GIS → **SOD** → **KAT** → **DIA**(7)
for both S-DFT and vdW-DF. However, the observation of the first two structures, RHO and GIS, was not reported in the HMeIm sequence (2). On the basis of our calculations, there was a priori no reason to discard RHO and GIS. If these two topologies could not be observed in the laboratory for any reason, then sequence (2) would be predicted in its entirety (except for the amorphous phase) by our calculations. We note, however, that the QTZ and DIA phases had very close positions in Figure 2a. Although DIA was less compact than QTZ, DIA had a lower energy and was included in sequence (7) instead of QTZ for this reason.

From Figure 2b we can see that the calculated sequence of HEtIm was more complex than that of HMeIm. Still using the ORS pathway, S-DFT calculations predicted the following phase sequence:
HEtIm (S-DFT): **RHO** → SOD → KAT → **QTZ**(8)

ANA, observed experimentally, is not included in this transformation sequence since its energy was about 100 meV above that of SOD. For vdW-DF calculations, all ZIFs considered, except CAG and possibly ANA, could be part of the transformation sequence:
HEtIm (vdW-DF): **RHO** → GIS → SOD (→ **ANA**) → NEB → KAT → DIA → **QTZ**.(9)

We note that the BH functional predicted that the energy of the ANA phase is only 10 meV higher than that of SOD, while the other five vdW-DFs predicted that the energy of the ANA phase is lower than that of SOD, as can be seen in Figure A1b. For this reason, we included ANA in parentheses in the sequence (9).

Within the assumed ORS pathway, our calculations indicated that the HMeIm and HEtIm transformation sequences involved many phases that were not noticed in the laboratory and cannot be ruled out without additional information.

### 2.2. CMD Quasi-Hydrostatic Pressure Breakpoint

In order to distinguish experimentally observed from hypothetical ZIFs, we calculated, here, an additional parameter (along with energy and porosity) characterizing ZIF structures, namely the quasi-hydrostatic pressure breakpoint. According to the theory of elasticity, the stability of a crystal under stress depends on the properties of the symmetric tensor of order four **A** defined as [21]:*A_ijkl_* = *C_ijkl_* + ½ (δ*_jl_ σ_ik_* + δ*_jk_ σ_il_* + δ*_il_ σ_jk_* + δ*_ik_ σ_jl_* − δ*_kl_ σ_ij_* − δ*_ij_ σ_kl_*),(10)
where the indices *i*, *j*, *k* and *l* refer to the three space coordinate axes, **C** is the elastic tensor of the crystal deformed by the applied stress tensor ***σ*** and δ_ij_ is the Kronecker delta. Since **C** and **A** have Voigt symmetry, they are conveniently represented as 6×6 symmetric matrices. For the crystal to be stable, the matrix **A** has to be positive definite, i.e., all its eigenvalues must be positive. This condition, called the Born condition, leads to six independent algebraic conditions or fewer for crystals with symmetries [22].

A few studies dealt with MOF stability under compressive, uniform pressure corresponding to *σ_ij_* = *P* δ*_ij_*, where *P* is the pressure. Using CMD, Bouëssel du Bourg et al. identified the pressure breakpoint of several ZIFs as an abrupt change in one of the crystal lattice parameters upon increasing the pressure in steps of 0.1 GPa [23]. For SOD-Zn(HIm)_2_ and CAG-Zn(HIm)_2_, the breakpoints were calculated to be 0.4 and 0.05 GPa, respectively. These values are in fair agreement with the measured pressures of 0.35 GPa for the amorphization of SOD-Zn(HIm)_2_ [24] and 0.12 GPa for the phase transition of CAG-Zn(HIm)_2_ into a phase called ZIF-4-I [25]. In the case of SOD-Zn(HIm)_2_, Ortiz et al. found, via CMD simulations, that the cubic crystal stability condition *A*_44_ ≥ 0 was not satisfied for *P* > 0.4 GPa, indicating that the crystal was subject to shear mode collapse [26]. More recently, Rogge et al. investigated the stability conditions determined from the **A** matrix and other methods for four MOFs of cubic or orthorhombic geometries [27].

To determine the quasi-hydrostatic pressure breakpoint at 300 K of the 18 ZIFs considered, we applied a pressure increasing by steps of 0.1 GPa between 0 and 0.4 GPa and then applied a pressure of 1 GPa. For each of the six pressure steps, the CMD simulations were run over 1 ns, as in [23]. Stepwise compression was performed using a Nosé–Hoover thermostat with a damping parameter of 0.1 ps. The time step was set to 0.1 fs. We considered two types of crystal structure. The first one was an infinite crystal built from the unit cell with periodic boundary conditions. For this purpose, the simulation box was made to fit the geometry of the unit cell. The second type of crystal structure consisted of a triple repetition of the unit cell in each direction in space. The clusters were placed in a rectangular box with periodic boundary conditions. In the latter case, the system was first relaxed under an isothermal-isobaric canonical ensemble (NPT) for 50 ps at 300 K and low pressure to allow the cluster and the simulation box to adjust to each other. For structures with α, β and γ angles of 90°, the results for both crystal structures were quite similar, if not identical. However, they differed in the other cases, although the pressure breakpoint obtained was almost the same. This can be seen in Figure 3, which shows the results of the CMD simulations for the time and pressure evolution of the nine topologies considered in this work with the HMeIm and HEtlm ligands. Except for DIA-Zn(HMeIm)_2_ and RHO-Zn(HEtIm)_2_, the volume of all structures decreased with increasing pressure. In most cases, the structure underwent sudden collapse at a certain pressure value, which we call the (quasi-hydrostatic) pressure breakpoint.

Examination of Figure 3 reveals that the three structures for each ligand that withstood the highest pressures with minimal volume change were precisely those that were observed experimentally, namely SOD, KAT and DIA for the HMeIm ligand (Figure 3a–c) and RHO, ANA and QTZ for the HEtIm ligand (Figure 3d–f).

In the absence of a clear criterion on the pressure breakpoint to distinguish observed from hypothetical ZIFs under the particular experimental conditions (reactor size, stirring frequency, ball masses), we took SOD-Zn(HMeIm)_2_ as a reference since, among the experimentally observed structures, it was the one with the lowest pressure breakpoint. Figure 4 shows different aspects of SOD-Zn(HMeIm)_2_ for some pressures. For P ≤ 0.2 GPa, there was no apparent change in the atomic arrangement. Symmetry and bond lengths were preserved. As the pressure increased to 0.3 GPa, the volume decreased by about 5%, as can be seen in Figure 3a. Compression in the horizontal direction was noticed, although the atomic patterns were globally preserved, and there was no apparent bond breakage. Although SOD had a cubic geometry at low pressure, fluctuations in the system, combined with the applied pressure, induced instability of the structure. When the pressure reached 0.4 GPa, the volume decreased by 10%. The geometrical patterns were very deformed, and many atomic bond breaks were observed. When the pressure reached 1 GPa, the structure was clearly destroyed. In this case, the pressure breakpoint was between 0.3 and 0.4 GPa. However, SOD became more brittle with the more complex HEtIm ligand, since the pressure breakpoint was below 0.2 GPa, as can be seen in Figure 3d. Therefore, we adopted the threshold value P_min_ = 0.3 GPa as a criterion to distinguish experimentally observed from hypothetical ZIFs.

From Figure 3, only SOD, KAT and DIA underwent a sudden volume change for P > P_min_ for the HMeIm ligand, whereas this condition was only met for RHO, ANA and QTZ (infinite crystal only in the latter case) for the HEtIm ligand. The remaining 12 structures underwent a significant volume decrease and structural changes at a pressure below P_min_. When this occurred, the structures were destroyed in the example of SOD-Zn(HMeIm)_2_ for P > P_min_ (Figure 4). However, the six structures supporting P_min_ did not behave in the same way when the pressure increases. Indeed, the volume of DIA-Zn(HMeIm)_2_ and RHO-Zn(HEtIm)_2_ was not affected up to 1 GPa; that of SOD-Zn(HMeIm)_2_ remained constant up to 0.2 GPa and then decreased stepwise; the volume of KAT-Zn(HMeIm)_2_ and ANA-Zn(HEtIm)_2_ decreased in small constant steps and then collapsed at 1 GPa; and the volume of the infinite crystal of QTZ-Zn(HEtIm)_2_ remained constant up to 0.2 GPa and then underwent fluctuations beyond this point but no collapse before 1 GPa.

The experimentally observed sequences (2) and (3) of ZIF appearance could, therefore, be deduced from the quasi-hydrostatic pressure breakpoint calculated by CMD with the additional rule, related to the ORS pathway, that successive ZIFs have increasingly lower energy and/or porosity.

## 3. Computational Methods

### 3.1. ZIF Atomic Models

We combined the six basic topologies of the reaction sequences (2) and (3), namely SOD, KAT, DIA, RHO, ANA and QTZ with the two ligands HMeIm and HEtIm to produce 12 ZIFs. We used the topologies of the CCDC database [28] which, however, are generally available with different ligands. The CCDC names of our starting structures were: ZIF-8 or OFERUNO3 (SOD), OFERUNO8 (KAT), OFERUN01 (DIA), MECWOH (RHO), MECWIB (ANA) and EHETER (QTZ). The two ligands HEtIm and HMeIm were incorporated in those structures using the Quantum ATK software [29]. For the six ZIFs actually observed in the mechanosynthesis process, the geometric structures so obtained were in agreement with those described in [15]. However, contrary to [15], we did not simplify the atomic arrangements by using symmetry considerations. The data for the GIS, CAG and NEB topologies were taken from [17], in which the HIm ligand was replaced by HEtIm and HMeIm to produce six additional ZIFs. We selected the three latter topologies in consideration of their small number of atoms in the unit cell and their experimentally verified stability at room temperature with the HIm ligand.

### 3.2. DFT Calculations

For the 18 ZIFs considered, we performed zero-Kelvin DFT energy minimization calculations to determine the equilibrium geometries and the corresponding total energies and porosities. We used the DFT code SIESTA since it allows efficient handling of structures containing several hundreds of atoms, and it comes with a flexible set of validated potentials for hybrid materials [30]. Periodic DFT calculations were carried out using the PBE exchange-correlation functional within the generalized gradient approximation (GGA) [19]. No symmetry was imposed on the structure in the calculation so that the cell may relax to alternative Bravais lattice types. Double-ζ polarization basis sets were used throughout. Pseudopotentials and basis functions for Zn, C, N and H were taken from the SIESTA database. In each case, the geometry was relaxed until the residual forces were smaller than 0.03 eV Å^−1^, with a stress tolerance less than 0.2 GPa. The Hartree and exchange-correlation potentials were evaluated using a real-space mesh with a kinetic energy cut-off of 250 Ry, while the Brillouin zone was sampled only at the Γ point 1×1×1. Baburin et al. also used SIESTA with similar settings to calculate the energy of a large variety of ZIFs with the HIm ligand [17]. We checked that the energies we calculated for a few ZIFs with the HIm ligand agreed to within 1% with their results.

It was demonstrated that vdW interactions play important role in the energetics of MOFs [31]. We, thus, carried out DFT optimizations, including these forces in the so-called vdW-DF framework, and we compared the results with those obtained using the PBE functional. The same calculation settings were used with and without vdW forces. All functionals, taking into account vdW interactions currently implemented in SIESTA, were tested, namely DRSLL [32,33], LMKLL [34], KBM [35], C09 [36], BH [20] and VV [37].

### 3.3. Packing Index

A convenient way to quantify the porosity of materials is the Kitaigorodskii packing index, which is defined as the volume of the molecules in the cell divided by the cell volume of the crystal. This index was calculated for the 18 DFT-optimized ZIF structures considered in this work using the PLATON software [38].

### 3.4. Molecular Dynamics Calculations

We performed QMD calculations using SIESTA at 300 K and ambient pressure for the six observed ZIFs of the sequences (2) and (3) to evaluate the effect of temperature on the cell volume as compared to zero-Kelvin calculations (see Table 1). We started from an optimized structure with a zero-Kelvin DFT calculation. At first, the structures were equilibrated in the NPT conditions. The temperature was controlled at 300 K using a Nosé–Hoover thermostat with a 0.1 ps temperature-damping parameter while the pressure was constrained at 0 GPa employing a Parrinello-Rahman barostat with a pressure-damping parameter of 1 ps. All QMD simulations were run over 500 ps with a time step of 1 fs. The volume and energy of the structures at 300 K were taken as the average over the last 100 ps of the simulation.

For the pressure breakpoint calculations, QMD was not an option due to the large computer time it requires. Instead, we performed CMD calculations which run much faster than QMD but likely at the cost of accuracy. Our CMD calculations were performed using the LAMMPS software [39], and atomic interactions between Zn, C, N and H were modeled using the reactive force field ReaxFF [40]. The advantage of ReaxFF over nonreactive valence bond potential is its ability to describe chemical reactions involving bond breaking and formation.

## 4. Conclusions

The aim of this work was to identify heuristic rules to predict the sequential appearance of mechanochemically synthesized ZIFs. For this purpose, we considered nine topologies of ZIFs combined with the ligands HMeIm and HEtIm for a total of 18 phases. For each ligand, only three topologies were observed during mechanosynthesis. As a working hypothesis, we first assumed that the ZIF phases underwent transformations to lower energy and lower porosity, following the Ostwald rule of stages. From the results of our DFT calculations, with and without van der Waals interactions, we concluded that it was not possible to account for the observations since several unobserved phases should have been included in the ZIF transformation sequences. Next, based on the fact that mechanochemistry involves high stresses due to ball impacts on the material covering the reactor walls, we turned to a third parameter characterizing ZIFs (in addition to energy and porosity), namely the quasi-hydrostatic pressure breakpoint. From CMD calculations, we found that mechanochemically generated ZIFs withstand higher pressures than hypothetical ZIFs. We used the pressure breakpoint of SOD-Zn(HMeIm)_2_, P_min_ = 0.3 Gpa, to distinguish observed from unobserved ZIFs. With the identification of the observed ZIFs, the additional condition that the energy and/or porosity of the structures decreases for each new ZIF generated allowed us to correctly predict the conversion sequences (2) and (3).

Although there is no direct relationship between unidirectional and random ball impacts on the material and the quasi-hydrostatic pressure considered in this work, both imply the general concept of crystal strength under stress. The relationship between the strength of the crystal phases during stress application and the experimental observation is probably that there are fewer high velocity ball impacts on the processed material (tail of the ball velocity distribution function) than low velocity impacts. Thus, the more robust crystalline phases will last long enough in the reactor to be observed. The less robust ones will appear but will undergo rapid transformations due to the more numerous low velocity impacts so that they can hardly be observed. According to this picture, several crystalline phases could coexist at any given time in the reactor, but with a small amount of less robust phases.

The next steps in this study would be to clarify the conceptual link between impact-induced stress and quasi-hydrostatic pressure. In addition, it would be interesting to establish the link between ligands, topology and the mechanical stability of crystal structures. Finally, the heuristic principles identified in this work could be used to predict the conversion sequences of other types of MOF to determine their range of validity. CMD calculations are relatively fast and could be used effectively for this purpose.

## Figures and Tables

**Figure 1 molecules-27-01946-f001:**
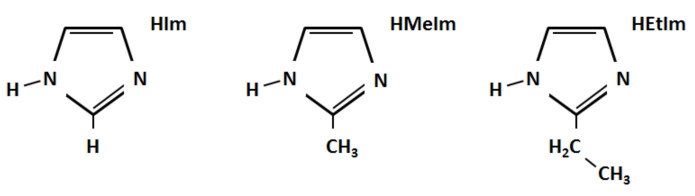
The ligands 2-imidazolate (HIm), 2-methylimidazolate (HMeIm) and 2-ethylimidazolate (HEtIm).

**Figure 2 molecules-27-01946-f002:**
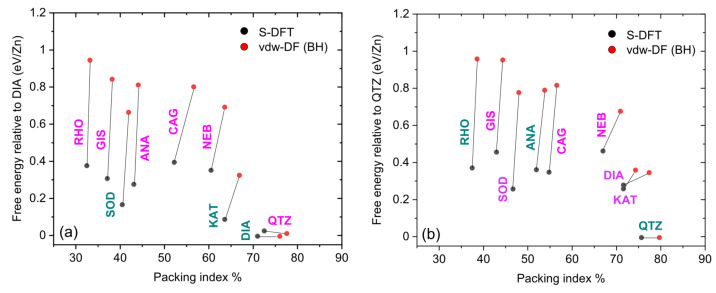
Relative energies for S-DFT (black dots) and vdW-DF (red dots) at 0 K, calculated using the BH functional [20]. (**a**) ZIFs with the HMeIm ligand. (**b**) ZIFs with the HEtIm ligand. The line segments connect the results of S-DFT and vdW-DF calculations for the same ZIF structure. The green labels denote the structures synthesized experimentally while the magenta labels denote hypothetical structures.

**Figure 3 molecules-27-01946-f003:**
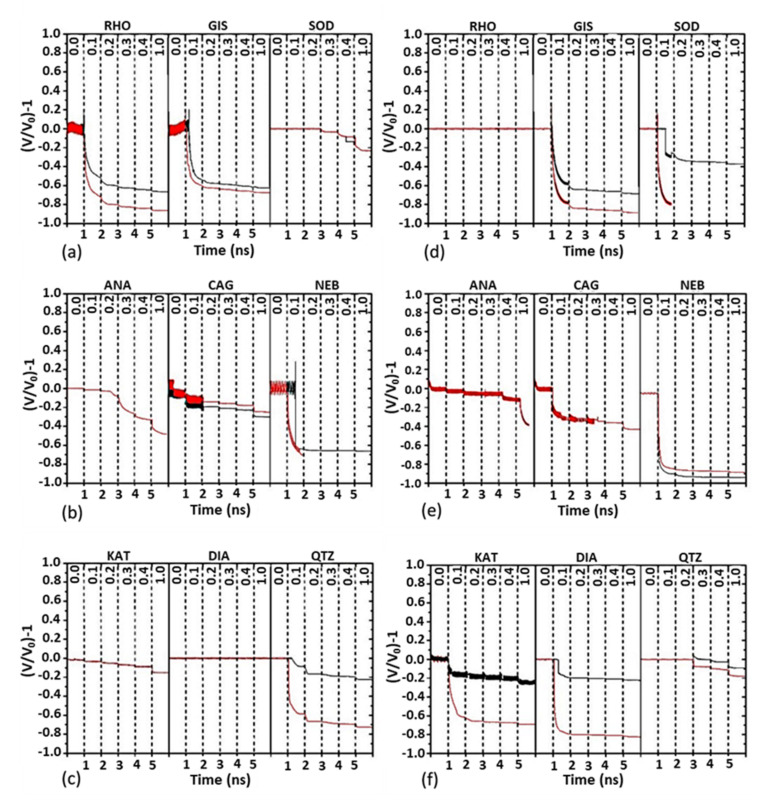
CMD calculation of the variation of the volume, normalized with respect to the average volume at zero pressure, as a function of time and pressure for the nine ZIF topologies considered in this work: (**a**–**c**) with the HMeIm ligand, (**d**–**f**) with the HEtIm ligand. Black: infinite structure, red: cluster. The pressure in GPa is indicated in the upper part of each figure.

**Figure 4 molecules-27-01946-f004:**
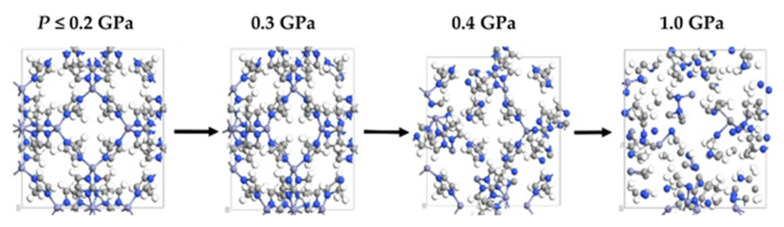
SOD-Zn(HMeIm)_2_ atomic structure at different applied, uniform pressures.

**Table 1 molecules-27-01946-t001:** Volume of the unit cells in Å^3^ for S-DFT and vdW-DF, with the BH functional [20], at 0 K and 300 K. The measured values are from [15]. Numbers in parentheses represent the relative deviation with respect to measurements.

	S-DFT 0 K	S-DFT 300 K	vdW-DF 0 K	vdW-DF 300 K	Measured
HMeIm sequence					
SOD	5083.36 (3.6%)	5108.36 (4.1%)	4792.24 (–2.3%)	4900.73 (–0.13%)	4907.124
KAT	4329.03 (1.8%)	4339.20 (2.0%)	4251.09 (0.0%)	4279.14 (0.7%)	4251.100
DIA	1908.97 (–0.4%)	1912.31 (–0.2%)	1914.38 (–0.1%)	1915.72 (0.0%)	1916.300
HEtIm sequence					
RHO	25547.22 (2.0%)	25,575.87 (2.1%)	24,720.69 (–1.3%)	24,990.42 (–0.2%)	25,046.300
ANA	19103.62 (2.3%)	19,202.95 (2.8%)	18,273.15 (–2.2%)	18,592.93 (–0.4%)	18,674.800
QTZ	817.55 (2.1%)	819.34 (2.3%)	769.65 (–3.9%)	788.75 (–1.5%)	800.860

## Data Availability

Data are available upon request from the authors.

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
