# Peer review of "Towards Predicting the Sequential Appearance of Zeolitic Imidazolate Frameworks Synthesized by Mechanochemistry"

_molecules, 2022, doi:10.3390/molecules27061946_

Round 1
Reviewer 1 Report
The manuscript entitled 'Towards predicting the sequential appearance of zeolitic imid-azolate frameworks synthesized by mechanochemistry by Vidal and co-workers deals with zeolitic imidazolate frameworks (ZIFs) generated by mechanochemical conversion.
Here are my comments:
I would add chromatographic separations as an additional filed of application for MOFs.
Please add references for mechanochemical transformations in recent literature. This is a very powerful technique, which is still under-explored:
T. Stolar, S. Grubešić, N. Cindro, E. Meštrović, K. Užarević, J. G. Hernández, Angew. Chem. Int. Ed. 2021, 60, 12727-12731.
K. J. Ardila-Fierro, J. G. Hernández, ChemSusChem 2021, 14, 2145-2162.
M. Haas, S. Lamour, S. B. Christ, O. Trapp, Communications Chemistry 2020, 3, 140.
S. Lamour, S. Pallmann, M. Haas, O. Trapp, Life 2019, 9, 52.
C. Bolm, R. Mocci, C. Schumacher, M. Turberg, F. Puccetti, J. G. Hernández, Angew. Chem. Int. Ed. 2018, 57, 2423-2426.
C. Bolm, J. G. Hernández, ChemSusChem 2018, 11, 1410-1420.
Author Response
We are delighted that our manuscript met the referee's requirements and thank him for the 6 additional references he provided. These have all been added to our manuscript to emphasize the great synthetic potential of mechanochemistry. We have also added reference [5] (Hu et al. 2014) regarding applications of ZIFs to chromatographic separations.
Reviewer 2 Report
The authors utilized comprehensive methods to investigate the sequential appearance of different structural topology when synthesizing zeolitic imidazolate frameworks by mechanochemical method. This study is of great importance in the community for future understanding these types of ZIF materials. All the experiments are well constructed and discussed, and the manuscript is well written and presented. I would be happy to recommend the publication without change.
I feel this work is well constructed and excuted, from the perspective of computational method. To be a very high standard, maybe corresponding experimental data can make this work much stronger. It do find a sepecial angle to understand the specific transformation sequence between these different topology. I am Ok with present methodology. No more suggestion here. The conclusion is highly consistent with evidence and arguments. The resolution of maintext figures (Figure 2-4) can be improved, and font size in these figures(Figure 2 and 3) can be a little bit bigger.Author Response
We are pleased that our manuscript has met the referee's requirements. Since the laboratory measurements were performed by another group, we, unfortunately, cannot reproduce their results in our paper. However, all the experimental results we used are available in reference [15] (Akimbekov et al. 2017) and in the supporting information of this same article. We have done our best to improve the readability of Figures 2, 3, and A1. Figure 4 was not retouched as we felt it was clear enough as it was. We hope that the figures will be to the satisfaction of the editor and the readers of Molecules.